# New Antimicrobial Strategies to Treat Multi-Drug Resistant Infections Caused by Gram-Negatives in Cystic Fibrosis

**DOI:** 10.3390/antibiotics13010071

**Published:** 2024-01-11

**Authors:** Viola Camilla Scoffone, Giulia Barbieri, Samuele Irudal, Gabriele Trespidi, Silvia Buroni

**Affiliations:** Department of Biology and Biotechnology “Lazzaro Spallanzani”, University of Pavia, 27100 Pavia, Italy; viola.scoffone@unipv.it (V.C.S.); giulia.barbieri@unipv.it (G.B.); samuele.irudal@iusspavia.it (S.I.); gabriele.trespidi01@universitadipavia.it (G.T.)

**Keywords:** alternative antimicrobial strategies, Gram-negative pathogens, cystic fibrosis, anti-virulence compounds, phage therapy, adjuvants, antimicrobial peptides, nanoparticles

## Abstract

People with cystic fibrosis (CF) suffer from recurrent bacterial infections which induce inflammation, lung tissue damage and failure of the respiratory system. Prolonged exposure to combinatorial antibiotic therapies triggers the appearance of multi-drug resistant (MDR) bacteria. The development of alternative antimicrobial strategies may provide a way to mitigate antimicrobial resistance. Here we discuss different alternative approaches to the use of classic antibiotics: anti-virulence and anti-biofilm compounds which exert a low selective pressure; phage therapies that represent an alternative strategy with a high therapeutic potential; new methods helping antibiotics activity such as adjuvants; and antimicrobial peptides and nanoparticle formulations. Their mechanisms and in vitro and in vivo efficacy are described, in order to figure out a complete landscape of new alternative approaches to fight MDR Gram-negative CF pathogens.

## 1. Introduction

Cystic fibrosis (CF) is a life-threatening autosomal recessive genetic disorder caused by mutations in the CF transmembrane conductance regulator (*CFTR*) gene, encoding a cAMP-regulated chloride channel expressed in the apical membrane of epithelial cells of different tissues. Inactivation of CTFR results in a complex, multisystem disease associated with the accumulation of thick, sticky mucus at the level of multiple organs, especially in the lungs, the gastrointestinal tract and pancreas [1]. Beside compromising pulmonary function, mucus deposition in the lungs creates an ideal microenvironment for bacterial colonization and recurrent infections, leading to inflammation and further progression of CF lung disease. *Pseudomonas aeruginosa*, *Burkholderia cepacia* complex (Bcc) bacteria, *Stenotrophomonas maltophilia*, *Haemophilus influenzae* and *Achromobacter xylosoxidans* represent the Gram-negative pathogens most commonly associated with CF lung infections [2].

Antibiotic discovery has tremendously contributed to the advancement of modern medicine and to the increase in life expectancy of people with CF (pwCF). However, the continuous use, misuse and exposure to antimicrobials have induced the rise in multidrug resistant (MDR) bacteria. In particular, people suffering from CF are repeatedly exposed to antibiotic treatments, leading to the development of MDR chronic infections [3]. Antimicrobial treatments involve the use of mono- or combinatorial-therapies that target essential metabolic processes, that exert a strong selective pressure and favor the appearance of resistant isolates [4]. 

In this scenario, identifying new antimicrobial strategies is mandatory. In this review, a collection of different innovative approaches to cope with MDR Gram-negative CF pathogens will be described. These include anti-virulence and anti-biofilm compounds, phage therapies, and the use of other alternative strategies as antibiotic adjuvants, antimicrobial peptides and nanoparticle formulations (Figure 1). The different approaches will be described from the point of view of their mode of action, including in vitro and in vivo assays that are useful to assess their efficacy. The review comprises all the innovative strategies that have received significant attention in the last five years, including their advantages and disadvantages. Overall, a comprehensive description of all the alternative approaches to fight MDR Gram-negative CF pathogens will be presented. 

## 2. Virulence and Biofilm Inhibition

Some advanced therapeutic strategies are based on preventing pathogen virulence, instead of killing them. These solutions are interesting alternatives to the use of antibiotics. One way to inhibit virulence is to target quorum sensing (QS), the cell-to-cell communication mechanism that allows bacteria to coordinate their behaviors. Among the many different processes controlled by QS, there is also the activation of defense mechanisms, such as the production of virulence factors (proteases, biofilm, immune-evasion factors and toxins) [5]. Both synthetic and natural anti-virulence compounds have been described and are herein reported. 

### 2.1. Synthetic Molecules

#### 2.1.1. Quorum Sensing Inhibitors

The *N*–acylhomoserine lactone (AHL)-dependent LasI/LasR circuit is at the top of the hierarchical QS cascade in *P. aeruginosa* [6]. Four small AHL analogs were designed by incorporating a tert-butoxycarbonyl Boc group in the amide and β-keto moiety. Assays performed using a LasR-based bioreporter strain revealed that the compounds compete with 3O-C12-HSL and decrease certain virulence traits (protease, elastase, pyocyanin production and extracellular DNA release) [7]. 

Halogenated furanone derivatives exhibit antagonistic activity against both LasR and RhlR receptors of *P. aeruginosa*: among these, one compound showed antibiofilm and antagonist activity against pyocyanin production and the expression of *lasB* [8]. This compound retained antimicrobial activity and hence could exert selective pressure for the development of resistance. On the other hand, a library of novel halogenated furanones was synthesized using a variety of palladium-catalyzed coupling reactions. They showed activity against biofilm formation in *P. aeruginosa*; one of them was demonstrated to be nontoxic and able to increase the survival of *Galleria mellonella* infected with *P. aeruginosa* [9]. 

Many different *Pseudomonas* quinolone signal (Pqs) modulators were synthesized and showed to reduce the production of virulence factors (such as pyocyanin and pyoverdine) and biofilm formation in *P. aeruginosa* strains derived from acute and chronic infections, without altering the cell viability [10]. In another study, 3-hydroxypyridin-4(1H)-one derivatives, bearing a 4-aminomethyl-1,2,3-triazole linker, were designed as anti-virulence agents against *P. aeruginosa* [11]. Among these compounds, one is a selective *pqs* inhibitor that blocks pyocyanin production and exhibits moderate biofilm inhibition activity. Moreover, it showed a synergistic effect in combination with antibiotics (ciprofloxacin and tobramycin) both in vitro and in vivo in a *Caenorhabditis elegans* infection model [11]. An integrated drug discovery campaign of new QS inhibitors (QSi) identified a molecule acting as an effective inverse agonist of PqsR which was able to abolish its activity and block downstream processes such as pyocyanin production [12]. The drug metabolism, pharmacokinetics and safety of the compound ensured it was suitable for pulmonary application. The efficacy of the QSi was demonstrated in a mouse model of *P. aeruginosa* mucoid lung infections. Moreover, a significant synergistic effect with tobramycin against *P. aeruginosa* biofilms was shown using a squalene-derived nanoparticles formulation [12]. 

In a drug-repurposing study using FDA (Food and Drug Administration)-approved drugs, two compounds, nitrofurazone and erythromycin estolate, were identified and demonstrated to be able to reduce both the expression of PqsE-dependent virulence factors and biofilm formation in *P. aeruginosa* PAO1, without affecting bacterial growth [13]. 

It is noteworthy that mRNA levels of Las QS-controlled genes are considerably lower under in vivo conditions compared to in vitro conditions [14]. On the other hand, the frequency of *rhlR* and *pqs* inactivating mutations is low in CF isolates. Drug repurposing studies showed that niclosamide, targeting the 3O-C12-HSL signaling process, has a low range of activity and its negative effect on *las* signal production does not induce a decrease in virulence factors [15]. Instead, clofoctol, acting as a competitive inhibitor of the signal receptor PqsR, displayed a broader QS inhibitory effect in CF isolates, with a reduction in the *pqs*-controlled factor pyocyanin [15]. These data support the development of *pqs* inhibitors as CF anti-virulence therapies and highlight the importance of assaying QS inhibitors on CF clinical isolates [16]. Results achieved to date suggest that anti-virulence therapies targeting RhlR or Pqs systems are more effective than LasR targeting strategies [17].

Regarding the other CF Gram-negative pathogens, one biogenic amine, called Tyramine, can inhibit the production of *N*-hexanoyl-homoserine signaling molecules (C8-HSL and C6-HSL), thus blocking the two QS systems of *B. cenocepacia* CepI/R and CCiI/R. This inhibition induces a reduction in many different virulence factors, like biofilm formation, extracellular polysaccharides, lipases and swarming motility [18]. Tyramine is not toxic in the model organism *G. mellonella* and showed high gastrointestinal absorption and the capacity to cross the blood–brain barrier. This molecule increased the efficacy of tetracycline against *B. cenocepacia* in a *G. mellonella* infection model [18]. Moreover, diketopiperazines are inhibitors of the AHL synthase CepI that decrease proteases, siderophores and biofilm production. These compounds are also active in the *C. elegans* infection model [19]. Using proteomic, site-directed mutagenesis and biochemical analyses, researchers identified a region close to the S-adenosylmethionine binding site critically involved in the inhibitor interaction [20]. 

MDR and biofilm production are phenotypes controlled by the QS system also through the diffusible signal factor (DSF) family. To identify new specific inhibitors, a library of sulfonamide-based DSF bioisosteres was prepared and tested against the QS-regulated phenotypes [21]. Several analogs enact interesting anti-biofilm activity against *S. maltophilia* and *Burkholderia cepacia* complex (Bcc) bacteria. Most compounds block DSF synthesis in *S. maltophilia*. In many cases, the compounds increase the action of colistin against both species [21].

All the described synthetic molecules able to inhibit QS are summarized in Table 1. 

#### 2.1.2. Biofilm Inhibitors

Some molecules showed activity against bacterial biofilms, blocking their formation or promoting their disassembly (Table 2). One example is the classical mucolytic drug, *N*-acetylcysteine (NAC), which, in different studies, showed the potential of inhibiting biofilm formation in *P. aeruginosa*, Bcc and *S. maltophilia* species and of disrupting mature biofilms of *P. aeruginosa* and *B. cenocepacia* [22,23,24]. The combination of NAC with antibiotics was investigated in CF pathogen-mixed biofilms, showing little effect on biofilms formed by *P. aeruginosa* and *A. xylosoxidans* [25].

Nitric oxide (NO) donors have been proposed as a dispersal treatment against *P. aeruginosa* biofilms, reducing the levels of the messenger cyclic-di-GMP [26]. Sodium nitroprusside, a NO donor and spermine NONOate (S150) can reduce biomass on a pre-established *P. aeruginosa* PAO1 biofilm [27]. Indeed, a two hour treatment with S150 induced biofilm disruption, without showing cytotoxicity. S150 has been proposed for preventing *P. aeruginosa* biofilm development or for treating existing biofilm; nevertheless, its molecular mechanism is still unclear. There are some concerns that repeated treatments with NO donors can lead to mutation, inducing a higher tolerance of biofilms to NO [27]. However, NO donors were potentially effective also against polymicrobial biofilms [28]. Moreover, to block *P. aeruginosa* planktonic cell release from biofilms, it is possible to combine an NO donor treatment with antibiotics [29]. 

Another interesting compound is the alginate oligosaccharide polymer Oligo G, which showed dose-dependent biofilm disruptive abilities. When used in combination with ampicillin and ciprofloxacin, it significantly decreased the minimal biofilm inhibitory concentration (MBIC) of the two antibiotics in *H. influenzae* [30]. Oligo G disrupted the biofilm structure of *P. aeruginosa* mucoid strains, improving the host immune system response and antibiotics activity [31]. Unfortunately, a first phase II study in CF adult subjects with *P. aeruginosa* infection demonstrated that the inhalation of Oligo G powder over 28 days did not significantly improve the forced expiratory volume in the first second (FEV1) [32]. 

Among heterocyclic corticosteroids, deflazacort and its synthetic precursor, the compound PYED-1, decreased *S. maltophilia* biofilm formation at sub-inhibitory concentrations. Hence, quantitative reverse transcription polymerase chain reaction (RT-qPCR) analysis showed that, in the presence of PYED-1, the expression of biofilm and virulence-associated genes is significantly decreased [33].
antibiotics-13-00071-t002_Table 2Table 2Synthetic molecules inhibiting biofilm.Synthetic Molecules

Biofilm InhibitorsBacteriaReference*N*-acetylcysteine (NAC)*P. aeruginosa*, Bcc and *S. maltophilia*[22,23,24]Nitric oxide donors, sodium nitroprusside*P. aeruginosa*[26,27]Oligo G *P. aeruginosa*, *H. influenzae*[30,31,32] Deflazacort, PYED-1*S. maltophilia*[33]


#### 2.1.3. Anti-Virulence Molecules

Active compounds could also block bacterial virulence factors other than biofilm, but involved in the establishment of the infection (Table 3). 

LasB, an extracellular metalloprotease, is an important virulence factor of *P. aeruginosa* implicated in pulmonary damage initiation. Its inactivation can be obtained by targeting QS-mediated activities or QS regulators (LasR, RhlR, PQS), blocking the production of autoinducers. Recent examples are the natural product derivatives psammaplin A and bisaprasin [34]. 

Pyocyanin toxin production could be decreased by blocking the QS systems. Examples of QSi are benzoxazolone derivatives that mimic AHL autoinducers [35]. One of these, B20, inhibited the pyocyanin production in *P. aeruginosa* PAO1, without interfering with bacterial growth. Expression of QS-promoting genes (*lasB*, *rhlA* and *pqsA*) was decreased in a dose-dependent manner after B20 treatment, suggesting that the reduction in pyocyanin is influenced by QS systems [35]. Since some QS genes are downregulated during CF infections, targeting genes directly involved in pyocyanin biosynthesis, such as *phzH*, *phzM* and *phzS*, could be considered a more appropriate approach [14]. 

Another possibility is to target siderophores, such as pyoverdine, which is essential for the establishment of *P. aeruginosa* infections. Gallium nitrate is considered an interesting antipseudomonal agent, since *P. aeruginosa* is unable to distinguish between Fe^3+^ and Ga^3+^, thus disrupting Fe metabolism. In this way, all the redox reactions critical for the virulence activities cannot occur. Recently, a study by Kang and co-workers demonstrated that Ga(NO_3_)_3_ treatment under iron-limiting conditions induces pyochelin biosynthesis: pyochelin binds and stores Ga^3+^ intracellularly, where it can interfere with cell functions [36]. Even if gallium has a bacteriostatic effect on bacteria, studies demonstrated that it exerts a low selective pressure, also at concentrations that interfere with bacterial growth [36]. 

MEDI3902 is a bivalent immunoglobulin IgG1 κ monoclonal antibody that targets PcrV and Psl of *P. aeruginosa*. PcrV is one of the major components of the Type III secretion system (T3SS) translocation apparatus and Psl is a structural component of the exopolysaccharide of *P. aeruginosa* biofilms. MEDI3902 reduced *P. aeruginosa* infections in animal models. The phase 1 clinical trial confirmed its safety in healthy subjects [37], but the phase 2 results were not satisfactory. However, other small molecules were studied as T3SS inhibitors: fluorothiazinon, which blocks the secretion of T3SS effectors ExoT and ExoY and protects mice from *P. aeruginosa* infections, is in a phase 2 clinical trials [38]. 

Regarding *H. influenzae*, IgA1 proteases are secreted virulence factors playing an important role in tissue invasion and immune response evasion. Screening of 47,000 molecules identified a hit compound inhibiting IgA1 activity [39]. Using a structure–activity relationship study, additional inhibitors were obtained, and two of them showed improved inhibition and selectivity for IgA protease [39]. 

Potentially, anti-virulence therapies can be combined with antibiotics to potentiate antimicrobial susceptibility. The non-mevalonate pathway inhibitor FR90098 was evaluated for its anti-virulence activity against *B. cenocepacia*, using in vitro and in vivo models [40]. FR900098, alone or in combination with ceftazidime, increased the survival of *G. mellonella* and *C. elegans*. Moreover, combined with ceftazidime, it produced a significant reduction in the biofilm formation of *B. cenocepacia*, and repeated exposures did not lead to a decrease in the activity [40]. 

### 2.2. Natural Compounds

#### 2.2.1. Quorum Sensing Inhibitors

Coumarin, derived from plants, natural spices and foods, has been previously described as a QSi with anti-virulence activity against *P. aeruginosa* [41]. Recently, a series of coumarin derivatives were evaluated, and the compound 4t was identified as an interesting biofilm inhibitor [42]. It inhibits QS systems but also competes as an iron chelator with pyoverdine, thus inducing an iron deficiency in *P. aeruginosa* [42].

Another natural compound is baicalin, one of the bioactive flavone constituents of *Scutellariae radix*: it is an AHL-based QSi active against *P. aeruginosa* and Bcc bacteria. In *P. aeruginosa*, baicalin reduces biofilm formation and the production of elastase, LasA protease, pyocyanin and rhamnolipids (*las* and *rhl* system-controlled virulence factors) [43]. Recently, it has been demonstrated that baicalin altered several virulence factors in *P. aeruginosa*, including the T3SS [44]. Baicalin reduced the toxicity of *P. aeruginosa* on mammalian cells and its virulence in a *Drosophila melanogaster* infection model. Moreover, baicalin treatment reduced the severity of lung damage and accelerated lung bacterial clearance. Regarding its molecular mechanism, data demonstrated that PqsR is required for baicalin’s effect on T3SS [44]. Unfortunately, in the case of baicalin as an antibiotic potentiator, it has been also demonstrated that, despite the fact that this QSi does not interfere with essential processes, repeated exposure to baicalin decreased the susceptibility of *B. cenocepacia* J2315 to the antibiotic-potentiating activity of the molecule [45]. 

Oridonin, identified in a natural products screening, can inhibit the motility, biofilm formation, protease production and virulence of *B. cenocepacia* and other *Burkholderia* species [46]. This compound, indeed, decreased the expression of the QS synthase-encoding genes, inhibiting BDSF and AHL production. Oridonin binds the RqpR regulator of the two-component system RqpSR, that controls the QS systems, to block the expression of the genes encoding the signal synthase. The compound can bind also the CepR of the *cep* AHL system, thus blocking its regulatory activities [46]. 

Another way to identify new active molecules is to screen environmental bacteria. For example, halophilic bacteria from marine samples were tested for their quorum-quenching activity. Among them, *Chromohalobacter* sp. D23 degraded C6 and C8-homoserine lactones and reduced biofilm formation in terms of total biomass and viability in *B. cepacia* cells [47]. When combined with chloramphenicol, the crude lactonase enzyme of *Chromohalobacter* sp. D23 increased the antibiotic susceptibility of *B. cepacia*. In general, *Chromohalobacter* sp. D23 reduced the QS-mediated synthesis of virulence factors (extracellular polymeric substances, extracellular proteases and hemolysins) [47]. 

Only celastrol, a pentacyclic triterpenoid compound found in *Tripterygium wilfordii* roots, has been demonstrated to decrease biofilm formation and disrupt established biofilms in *S. maltophilia*. The anti-virulence effect of celastrol is exerted by decreasing protease production and interfering with *S. maltophilia* motility [48]. 

#### 2.2.2. Biofilm Inhibitors

Many different natural compounds were investigated for their biofilm inhibition and eradication potential against *P. aeruginosa*. Among these, a plant extract of *Dioon spinulosum*, a variety of giant Cycads, induced a reduction in biofilm formation of *P. aeruginosa* isolates both in vitro and in vivo rat models [49]. 

Specialized motility allows Gram-negative bacteria to spread and form biofilms. Plant-derived triterpenes, analogs of the oleanolic acid, were evaluated against *P. aeruginosa* clinical isolates using biofilm formation assays and swarming assays. One of these inhibited the swarming of different *P. aeruginosa* isolates and of *B. cenocepacia* [50]. This compound potentiated antibiotic activities (tobramycin and colistin). qPCR data suggested that it altered the expression of genes involved in type IV pili regulation [50]. 

Among the plant essential oils, immortelle (*Helichrysum italicum*), *Origanum majorana*, thyme and citrus showed valuable biofilm inhibition activity against *P. aeruginosa* and *H. influenzae* [51,52,53,54]. 

Another study focused the attention on ceragenins (CSAs), cationic steroid antibiotics derived from bile acid modified to yield an amphiphilic morphology. All the CSA derivatives showed significant biofilm inhibitory activity against *A. xylosoxidans* [55].

An alternative strategy is to screen environmental resources for active compounds. For example, enrichment cultures of five marine resources were analyzed using sequence-based screening coupled with deep omics analyses to search for enzymes with antibiofilm characteristics. The supernatant of the stony coral caused a 40% reduction in *S. maltophilia* biofilm formation [56]. Further investigations on the culture’s metagenome and proteome indicated an important group of metalloproteases responsible for this activity [56]. Ascorbic acid (vitamin C) is an antioxidant and a micronutrient that sustains immune system functions. In *S. maltophilia,* vitamin C can inhibit biofilm formation in a concentration-dependent manner [57]. Also, the extract of *Allium stipitatum* reduces biofilm viability and structure in *S. maltophilia* [58]. 

Studies regarding biofilm inhibition are focused on glycosyl hydrolases involved in the degradation of exopolysaccharide, one of the major components of the biofilm matrix. The glycosyl hydrolase PslG of *Pseudomonas fluorescens* can exert an anti-biofilm activity on a series of *Pseudomonas* strains, disassembling preformed biofilms [59].

Natural compounds showing anti-QS or antibiofilm activity are listed in Table 4.

## 3. Phage Therapy

Bacteriophages (phages) are viruses able to specifically infect and kill bacteria. In this way, their use as an alternative to antibiotics to treat MDR pathogens emerged a few years ago [60], although their first successful clinical use was achieved in 2016 [61]. So far, phage therapy has been allowed in the U.S.A. for compassionate use, e.g., to treat pwCF with pulmonary MDR infections, while, in Europe, bacteriophages have been classified as medicinal products [62,63]. These limitations are mainly due to the absence of a gold standard in phage preparation and administration and to a gap in the knowledge regarding their efficacy and safety, but also to the difficulties in conducting clinical trials.

However, in the last five years, many researchers have focused their attention on this alternative solution applied to Gram-negative CF pathogens.

### 3.1. H. influenzae

Regarding *H. influenzae*, one study published in 2019 reported the use of a mouse model of infection to test the efficacy of phage therapy [64]. First of all, the dose to be administered was set up. The problem of the restricted host range and lyses of only few strains within the same species was solved by the preparation of a polyvalent cocktail, which contained four phages active against different strains belonging to the same bacterial species [64]. The authors explain this as a result of phage adsorption to several receptors such as lipopolysaccharide (LPS), protein-LPS combination, outer membrane proteins, enzymes localized on outer membrane and selective transport protein(s) [64].

### 3.2. A. xylosoxidans

An interesting case report about the use of phages to treat a 17 year old girl suffering from MDR *A. xylosoxidans* infection was published in 2018: the patient was given a therapeutic phage cocktail (two phages isolated from different wastewater samples in Germany) daily via inhalation and per os for 20 days for four times. Her scheduled IV antibiotic therapy was administered 6 months after phage therapy. The patient’s conditions and lung function significantly improved, indicating that treatment of CF patients with phages can allow a reduction in antibiotic use and the need for hospitalization [65].

Similarly, a successful use of phage therapy and antibiotics (cefiderocol, meropenem/vaborbactam) was reported in 2020 to treat a 10-year-old girl infected with a pan-drug resistant (PDR) *A. xylosoxidans* [66].

The first case of phage therapy for a recurrent PDR *A. xylosoxidans* infection in a lung-transplanted patient was reported in 2021 [67]. Here, the therapy was complicated because of the presence of one *Achromobacter* strain carrying a stop mutation in a gene encoding a phage receptor. However, no re-colonization occurred after two rounds of therapy [67].

Recently, Cobián Güemes and collaborators reported the isolation, characterization and production of six distinct lytic *Achromobacter* phages, identified thanks to a comprehensive phylogenetic analysis [68]. 

### 3.3. B. cepacia Complex

Regarding the genus *Burkholderia*, in 2021, Godoy et al. [69] reported the genome sequence of a P2-like phage of *Burkholderia gladioli*, a ubiquitous Gram-negative with a recognized ability to infect pwCF. The phage was isolated from soil and sequenced to characterize its main features, in order to advance knowledge on alternative treatments for *B. gladioli* infections.

The following year, a broad host range Podovirus was isolated and characterized; although it could lysogenize the Bcc bacteria, it had some good characteristics for phage therapy. For instance, it had a high global virulence index, a high infection efficacy and stability [70]. The same aspect was considered by Lauman and Dennis, who developed novel metrics that showed a strong inverse correlation between lysogen formation and antibacterial activity, thus confirming that certain lysogenic phages may be therapeutically effective [71].

A phage–antibiotic synergy approach has been used to enhance the killing of *B. cenocepacia*, by combining a lytic phage isolated from raw sewage with meropenem, ciprofloxacin and tetracycline [72].

In the same year, the “phage steering” approach was shown to decrease *B. cenocepacia* virulence and to increase its antibiotic susceptibility [73]. Indeed, phage-induced resistance has been demonstrated to be responsible for the alteration of the LPS, since random mutations occur in the receptors, giving rise to truncated forms of LPS. In turn, this enhances bacterial sensitivity to immune components and to membrane-associated antibiotics, such as polymyxin B and colistin. In the same study, the phage treatment strategy termed “anti-virulence” was applied; here, phage resistance is achieved when a bacterial virulence factor is used as a receptor and, when it is modified or deleted, a reduction in virulence occurs [73].

Unfortunately, a case of clinical failure of nebulized phage therapy in a lung transplanted pwCF infected with MDR *Burkholderia multivorans* highlighted the limitations, unknown aspects, and challenges of this alternative approach for resistant infections [74]. Indeed, a reduction in phage viability due to the nebulizer used was observed; older *Burkholderia* isolates tended to be more antibiotic-susceptible and phage-resistant; and the rapid decline of the patient may have been associated with phage administration. However, these results came from a single very ill patient, making any conclusions about the effect of phage therapy controversial.

### 3.4. S. maltophilia

A lot of *S. maltophilia* targeting phages have been isolated and characterized in the last three years. An initial one, AXL3, was isolated from soil and shown to be able to infect five isolates. It is a member of the Siphoviridae family, which uses the type IV pilus as receptor [75].

Similarly, AXL1 was isolated from the same source. Functional genomic analyses revealed the presence of a dihydrofolate reductase enzyme conferring resistance to the antibiotic combination trimethoprim–sulfamethoxazole, providing an example of phage-encoded antibiotic resistance [76].

Another one, DLP3, showed a broader host range and was lysogenic, but at the same time demonstrated excellent therapeutic potential because of its broad host range, its ability to infect host cells through the *S. maltophilia* type IV pilus and its lytic activity in vivo [77]. In this way, the authors proposed to eliminate the temperate lifecycle by using genetic techniques.

In 2021, the vB_SmaS_BUCT548 phage was isolated from the sewage of a Beijing hospital. It possesses a relatively wide host range, a short incubation period and strong lytic activity [78].

In 2022 another phage, vB_SM_ytsc_ply2008005c, was isolated in the same manner and characterized [79].

A Podoviridae phage, named BUCT598, was isolated from the same source. Its characterization is extremely helpful to understand phage adaptation and evolution [80].

In addition, BUCT603 was classified as a new member of the Siphoviridae family. It showed a broad host range and used the TonB protein as a receptor. Its efficacy was assessed in a mouse model, suggesting its great potential as a candidate for the treatment of *S. maltophilia* infection [81].

The bacteriophage BUCT700 uses the type IV fimbrial biogenesis protein PilX as a receptor. It showed excellent thermal stability and pH tolerance, and it was able to maintain a high titer during long-term storage. It also showed good in vivo activity, increasing the survival rate of *S. maltophilia*-infected *G. mellonella* larvae [82].

The accumulation of phage resistant mutations and the acquisition of the phage defense systems have been analyzed by Zhuang and co-workers [83], who evaluated the impact of (pro)phages in *S. maltophilia* clinical isolates. Highly variable parts of the genome were detected. The pan-immune system maps of these strains against phage infections revealed a co-evolutionary dynamic between bacteria and phages. Furthermore, 310 prophage regions, as well as six viral defense systems, were identified [83].

Finally, the Myoviridae CM1 phage was recently isolated and characterized, revealing common features with other *S. maltophilia* phages [84]. 

### 3.5. P. aeruginosa

Many papers in the last few years reported the characterization of phages to treat MDR *P. aeruginosa* and investigated strategies to exploit them in therapy, including the use of combinations with antibiotics in powder formulations [85,86], the purification of lysins [87], the set-up of a zebrafish model to check the phage effect [88], the evaluation of antagonistic effects of antibiotics towards phages by lysis-profile assays [89], the evaluation of the efficacy of environmental phages against *P. aeruginosa* biofilms [90], the identification of broad host range lytic phages [91], the assessment of cocktails to be used against MDR strains [92,93], the exploration of phages engineered with anti-CRISPR genes to render *P. aeruginosa* unable to replicate and infect [94], the combined use of liposomes and a phage-cocktail to improve the antimicrobial effect of single treatments [95] and the use of genome sequencing, comparative genomics and lytic activity screening to improve phage characteristics [96].

In 2022, a randomized, placebo-controlled, double-blind study was published, in which a single dose of intravenous phage was administered to 72 clinically stable adult pwCF with *P. aeruginosa* airway colonization [97]. The final goal was to fulfill the main gaps in the knowledge about the efficacy of phage therapy, optimal frequency, dosage, and duration of phage administration; the standardization of references to predict phage susceptibility; the evaluation of the frequency of the emergence of phage resistance; the role of the human immune system in phage efficacy; and the understanding of the safety profile of phage therapy.

The need for further clinical trials, also at the pediatric level, has been highlighted by Hahn and colleagues [98], who reported the details of two pwCF with PDR *P. aeruginosa* treated with personalized inhaled bacteriophage therapy. The authors conclude that phages have the potential to be used as direct therapy to fight not only *P. aeruginosa* lung infection, but also other hard-to-treat bacterial pathogens to increase antibiotic treatment of pulmonary exacerbations, or potentially eradicate initial pulmonary colonization in young children with CF. In this way, properly designed trials are urgently needed to assess the safety and efficacy of phage treatments to be used for all pwCF [98].

Finally, very recently, an obligately lytic phage was combined with cationic Zinc (II) porphyrin founding synergy against *P. aeruginosa* in an in vitro lung model and showing greater protection of lung cells than with either treatment alone [99].

## 4. Antibiotic Adjuvants

Adjuvant molecules can be divided into three main classes, based on their mechanism of action: outer membrane (OM) perturbing agents that facilitate the entry of antibiotics and increase their intracellular concentration; efflux pump inhibitors, which prevent the extrusion of the antibiotics from the bacterial cell; and β-lactamase inhibitors, blocking the enzymatic degradation of β-lactams exerted by periplasmic β-lactamases (Figure 2). However, sometimes these classes overlap, since certain molecules target multiple resistance determinants, as in the case of OM perturbing agents that, depolarizing the inner membrane, often also block efflux pump activity through proton motive force dispersion. Within the last five years, research into new antibiotic adjuvants has been mainly focused on OM perturbing agents against the major CF pathogen, *P. aeruginosa*; for this reason, this part of the review will be mostly dedicated to this topic.

### 4.1. Outer Membrane Perturbing Agents

#### 4.1.1. Polymyxin-Derived Adjuvants

Historically, the first molecule characterized as OM permeabilizer was polymyxin B nonapeptide (PMBN) in the 1980s, which, differently from the parent molecule polymyxin B (PMB), lacks antimicrobial activity but shows a potent synergistic effect in combination with hydrophobic antibiotics [100]. A PMB derivative showing synergistic activity with several clinical antibiotics against *P. aeruginosa* was reported by Domalaon and colleagues [101], testing the efficacy of dilipid polymyxin derivatives. In this study, the derivative 1 was identified as a potential adjuvant, although it showed an efficacy and toxicity profile comparable with the extensively studied PMBN. In another work, guanidinylated polymyxin derivatives were synthesized, substituting the L-γ-diaminobutyric acid amines with guanidines to decrease antibacterial activity and improve the OM perturbing effect. This modification led to guanidinylated PMB and colistin that were able to synergize with rifampicin, erythromycin, ceftazidime and aztreonam against MDR Gram-negative pathogens, including *P. aeruginosa* strains [102]. The two derivatives were more effective than PMBN as adjuvants but, unfortunately, none of them showed decreased cytotoxicity in vitro compared to PMB, thus requiring a further chemical optimization.

#### 4.1.2. Polyamine Derivatives

Natural polyamines as spermine, spermidine, squalamine and ianthelliformisamine are polycationic molecules known to exert antibiotic enhancer efficacy against different Gram-negatives, including *P. aeruginosa* [103,104]. However, these molecules are effective only at very high concentrations, limiting their development as adjuvants. Given the potential of polyamines, in recent years, a multitude of derivatives were synthesized and tested to find optimized candidates for future development in clinics. Among them, the most promising derivative was the polyaminoisoprenyl molecule NV716, showing remarkable efficacy as an OM permeabilizer, and significantly decreasing the MIC values of poorly permeable antibiotics, such as doxycycline, chloramphenicol, rifampicin and linezolid, against *P. aeruginosa* strains, including clinical isolates. Interestingly, no stable resistance was reported for NV716 after up to 50 serial passages in liquid culture and, when used in combination, it decreased the frequency of resistance to antibiotics [105]. Moreover, an antibiotic adjuvant effect was observed against sessile bacteria within structured biofilms and against intracellular bacteria [105,106]. NV716, additionally, combines antibiotic adjuvant activity with other useful antibacterial characteristics such as anti-virulence properties, reducing quorum sensing-related gene expression and decreasing the fraction of persister cells in stationary phase cultures when combined with ciprofloxacin [106]. In view of a clinical use of NV716 as CF antimicrobial therapy, an inhalable formulation doxycycline/NV716 was successfully developed, which was potentially suitable for *P. aeruginosa* lung infection treatment [107]. 

Other promising molecules active against *P. aeruginosa* include norspermidine and indoglyoxylamide polyamine derivatives. NAda and NDiphe are norspermidine derivatives incorporating cyclic hydrophobic moieties. These modifications led to decreased cytotoxicity and increased adjuvant properties, thanks to weak OM permeabilization and efflux pump inhibition [108]. 6-bromoindolglyoxylamido-spermine is a modified marine natural product with a comparable mechanism of action, effective as a potentiator of doxycycline against *P. aeruginosa* [109]. Due to its relevant cytotoxicity, several modifications of the original molecule were performed to improve its safety, testing different substituents on the indole ring, changing the length of the polyamine core or adding conjugated molecules [110,111,112,113,114]. 

#### 4.1.3. Cationic Peptides

Cationic antimicrobial peptides kill bacteria mainly through membrane disruption and consequent lysis. However, natural peptides possess several drawbacks, such as protease instability, high production costs and cytotoxicity at high concentrations [115]. For these reasons, optimized synthetic peptides are being studied, focusing also on those devoid of intrinsic antimicrobial activity but with an interesting OM perturbation potential. Among them, promising results were obtained with short lipopeptides, such as the proline-rich monoacylated heptapeptide C_12_-PRP, which potentiated the effect of minocycline and rifampin against *P. aeruginosa* MDR clinical isolates, with no detectable cytotoxicity [116]. Dilipid ultrashort cationic lipopeptides, instead, showed good adjuvant activity and low cytotoxicity in combination with chloramphenicol, in case of lysine-based peptides [117], or novobiocin and rifampicin, in case of arginine-based peptides [118]. 

A better characterized short peptide is the cathelicidin derivative D-11, enhancing the efficacy of several antibiotics against *P. aeruginosa*, particularly the macrolide azithromycin, one of the main CF antibiotics, through the permeabilization of the OM and the dissipation of the proton motive force. This combination was active against biofilms and in a mouse model of infection, highlighting its potential against CF infections [119]. 

A different approach, beside short peptides synthesis, is the use of macromolecular potentiators, composed of antimicrobial peptides bound to a polymeric scaffold. These molecules have some advantages compared to the monovalent AMP, such as increased activity at lower concentrations and enhanced stability. One example is the multivalent peptide construct WP40, characterized as an excellent OM permeabilizer able to significantly increase the efficacy of hydrophobic antibiotics against *P. aeruginosa*, with a decreased cytotoxicity compared to the monovalent peptide [120]. 

High molecular weight amino acid polymers are also studied as adjuvants against *P. aeruginosa*. In particular, poly-L-lysine was already reported to be a mucolytic and biofilm perturbing agent, but further characterization established this molecule as imipenem, ceftazidime and aztreonam potentiator effective ex vivo, in a 3D model of human primary bronchial cells, against clinical strains [121].

#### 4.1.4. Antibiotic Conjugates

Polybasic antibiotics have the potential to destabilize LPSs and consequently disrupt OM permeability barrier of Gram-negatives. For instance, when used at high concentrations, tobramycin exerts bactericidal activity against *P. aeruginosa* primarily via a membrane-associated mechanism, instead of inhibiting protein synthesis [122]. For this reason, tobramycin conjugates were synthesized, resulting in potent OM permeabilizers with adjuvant properties and, importantly, no cytotoxicity. Tobramycin was conjugated with different antibiotics, such as ciprofloxacin, levofloxacin and rifampicin, obtaining potent antibiotic enhancers, in particular in combination with fluoroquinolones, tetracyclines and hydrophobic molecules, such as rifampicin [123,124,125]. Other promising hybrid molecules include the tobramycin conjugate with efflux pump inhibitors [126] and the chelating agent cyclam [127]. Interestingly, the latter conjugate was able to revert β-lactam resistance, not only by modifying OM permeability but, probably, also by indirectly decreasing OprD porin down-regulation [127]. Recently, dimeric and trimeric tobramycin [128,129] and chimeric tobramycin-based adjuvants [130] were synthesized, testing new conjugates composed of up to three membrane-active compounds.

### 4.2. Efflux Pump Inhibitors

Antibiotic resistance in Gram-negative pathogens as *P. aeruginosa* is mainly mediated by the expression of efflux pumps. These are typically grouped into five different families with different substrate specificities [131,132]. The clinical development of broad-spectrum efflux pump inhibitors would be desirable, but the inherent cytotoxicity of these molecules limits the feasibility of this approach. Instead, the characterization of inhibitors of specific efflux pumps could mitigate this problem. To this end, derivatives of the MexXY-OprM inhibitor berberine were synthesized and tested in combination with tobramycin against different *P. aeruginosa* strains. Moreover, the interaction of these molecules with MexY variants was analyzed in silico, allowing for future molecule optimization [133]. 

Recently, another class of efflux pump inhibitors, namely halogenated indoles, was synthesized and the representative molecule, 4F-indole, was characterized as a possible MexXY-OprM inhibitor. This molecule showed an aminoglycoside-enhancing effect, similar to berberine derivatives, but with a different mechanism of action that relied on indirect inhibition of the efflux pump, through the activation of the PmrAB two-component system [134].

### 4.3. β-lactamase Inhibitors

β-lactam antibiotics are essential for the treatment of Gram-negative CF infections. The continuous isolation of new β-lactamase encoding genes challenges the effectiveness of these treatments, leaving clinicians with a few therapeutic options. Among them, class B or metallo-β-lactamases (MBLs) are a major problem, given that no inhibitor is currently available against these enzymes. Since divalent zinc ions are the essential cofactor for MBLs, tris-picolylamine-based zinc chelators were synthesized as potential inhibitors, and one compound was the best candidate for further studies. Indeed, it was effective in reverting meropenem resistance against an MBL-producing *P. aeruginosa* strain and, importantly, it was well tolerated in mice [135]. 

In another recent work, six previously characterized zinc chelators were tested in vitro and in a *G. mellonella* model of infection against carbapenem-resistant *S. maltophilia*, identifying TPEN and nitroxoline as promising synergists with meropenem [136]. 

Due to the expression of Pen-like and AmpC serine β-lactamases, Bcc bacteria are naturally resistant to β-lactams, including the recently approved ceftazidime/avibactam combination [137]. To overcome this problem, a novel combination of avibactam with piperacillin was tested against a panel of Bcc and *B. gladioli* isolates, restoring susceptibility to β-lactams of 99% of the strains [137] and providing a valuable and safe alternative treatment. 

In addition, the novel combinations imipenem/relebactam and cefepime/zidebactam were tested against Bcc and *S. maltophilia*, resulting effective at low concentrations against most isolates [138,139].

### 4.4. Other Adjuvants

Besides the above-described molecules, it is possible, in the literature, to find other compounds that are able to potentiate antibiotics against Gram-negative CF pathogens, through different or not-well-characterized mechanisms of action. Among them, there is one FDA-approved molecule reported to increase polymyxin efficacy against *P. aeruginosa*, namely niclosamide, an anthelmintic drug. Niclosamide and its derivatives were characterized as potent adjuvants of polymyxin B and colistin, able to completely resensitize resistant strains, by inhibiting the expression of a gene cluster involved in LPS modification, and to decrease the insurgence of resistant mutants [140,141]. Interestingly, this compound was previously described as QSi, highlighting the importance of screening already characterized molecules for new biological effects (Section 2.1.1; Table 1). 

Biofilm-controlling agents, such as aspartic acid and succinic acid, were tested as ciprofloxacin adjuvants for prophylactic or treatment approaches against *P. aeruginosa* polymicrobial biofilms. Both were very effective in preventing the formation of or in aiding in the eradication of biofilms in vitro, although only aspartic acid showed a safe toxicity profile [142]. 

Vitamin E was characterized as an antibiotic adjuvant targeting a *B. cenocepacia* extracellular mechanism of resistance exerted through the activity of the lipocalin protein BcnA. Indeed, this protein can capture different antibiotics, preventing their interaction with bacterial cells. Exploiting the high affinity of vitamin E for BcnA, an effective adjuvant, active in vitro and in vivo, was developed [143]. 

As described in Section 2.1.2, NAC activity against bacterial biofilm was assessed. Moreover, combinations of NAC and antibiotics were tested against *B. cenocepacia* [144], *S. maltophilia* [145], *A. xylosoxidans* [23,146] and *P. aeruginosa* [25], which were capable of increasing the efficacy of antibiotics such as ciprofloxacin, colistin and tobramycin against planktonic bacteria. The mechanism of action of NAC adjuvant activity is still unknown, although it is probably multifactorial, involving NAC proteolytic properties, the inhibition of exopolysaccharide production and the impairment of the cysteine metabolism [147].

## 5. Antimicrobial Peptides

Antimicrobial peptides (AMPs) are a diverse group of short (10–50 amino acids long), cationic and amphipathic peptide molecules that are produced as important components of the innate immune system by almost all organisms. Besides acting as a first line response to invading pathogens, they play a key role in the modulation of the immune response and the inflammatory process [148,149,150].

Despite an enormous diversity in their amino acid sequence, AMPs share biophysical properties that confer them broad-spectrum antibacterial activity: their net cationic charge enables direct interaction with the anionic components of the bacterial surface, while their hydrophobic part contributes to the penetration and perturbation of lipid bilayers, eventually leading to the alteration of membrane permeability and the killing of the target bacterium [151]. Notably, some AMPs do not rely on a direct membranolytic mechanism, but pass through the bacterial cytoplasmic membrane, without necessarily disrupting it, and interfere with essential processes like DNA and protein synthesis, inhibition of cell division and cell wall biosynthesis [152,153]. 

### 5.1. Natural Antimicrobial Peptides

The rhesus theta-defensin-1 (RTD-1), a natural macrocyclic AMP expressed in leukocytes of Old World primates, has been proposed as a promising potential therapeutic agent for CF airway infections. This molecule exhibited a rapid in vitro bactericidal activity against mucoid, non-mucoid and MDR clinical isolates of *P. aeruginosa*, showing no cross-resistance to colistin [154,155]. Importantly, the in vitro activity of the compound correlated with its in vivo efficacy. When administered via nebulization in a CFTR F508del-homozygous murine model of chronic *P. aeruginosa* pulmonary infection, RTD-1 effectively decreased the lung infection burden. The in vivo antibacterial activity was coupled to a potent anti-inflammatory ability, which was exerted through the disruption of NF-κB signaling and the consequent inhibition of inflammasome formation [156]. The combination of the antimicrobial and anti-inflammatory properties of RTD-1 makes this compound a promising therapeutic approach for the treatment of CF lung infections [157]. Pharmacokinetics, safety and tolerance studies demonstrated intrapulmonary safety, tolerability and stability of RTD-1, supporting the possibility of the aerosol administration route [157]. Notably, the macrocyclic structure of RTD-1 confers drug stability and resistance to cleavage by the highly abundant proteases present in the CF sputum. This is particularly relevant if we consider that proteolytic degradation and reduced activity in the pH and salt concentrations found under physiological conditions represent the major limitations to the use of peptide drugs in vivo [158].

These issues must be taken particularly into account when we consider the lung environment of pwCF, which is characterized by an acidic pH, high concentration of salts and the presence of viscous mucus. Naturally occurring cathelicidin peptides with in vitro antibacterial activity against clinical CF isolates of *P. aeruginosa*, *S. maltophilia* and *A. xylosoxidans* exhibited minimal killing capacity against *P. aeruginosa* when tested in CF sputum [159].

### 5.2. Strategies to Overcome the Drawbacks of the Use of Antimicrobial Peptides In Vivo

Different strategies have been developed to overcome the susceptibility to protease degradation and the hemolytic and cytotoxic properties associated with the use of AMPs in vivo. These include the shortening of peptide lengths, the chemical modification of natural AMPs and the design and synthesis of natural-derived or non-natural (peptidomimetics) peptides. 

Two cathelicidins of bovine origin (BMAP-27 and BMAP-28) showed rapid and potent in vitro bactericidal and anti-biofilm activity against CF isolates of *S. aureus*, *P. aeruginosa* and *S. maltophilia*, when tested under experimental conditions mimicking the physical and chemical properties of the CF lung environment. Notably, they exhibited an activity higher than tobramycin [160,161]. 

With the aim to obtain molecules more feasible for a clinical use, *N*-terminal shortened fragments of these peptides were produced. Despite maintaining an in vitro activity similar to that of parental compounds, the new truncated derivatives caused acute toxicity when intratracheally administered to mice lungs. Among them, the 1–18 *N*-terminal fragment of BMAP-27 (BMAP18), selected for its good antimicrobial potential and reduced pulmonary toxicity, lost its in vivo antibacterial activity due to its rapid degradation by pulmonary proteases in a murine *P. aeruginosa* lung infection model [162].

Enantiomerization represents an appealing approach to solve the intrinsic stability problem of AMPs. This strategy was successfully applied to synthesize a derivative of the frog skin-derived antimicrobial peptide Esc(1–21): the substitution of two amino acids with the corresponding D-enantiomers increased biostability, lowered cytotoxicity and improved antibiofilm and in vitro and in vivo antibacterial activity against *P. aeruginosa* [163,164].

Similarly, in order to enhance peptide resistance to pulmonary proteases, the all-D-isomer of BMAP18 was synthesized (D-BMAP18). D-BMAP18 retained relevant antimicrobial activity against planktonic forms of clinical CF isolates of *P. aeruginosa* and *S. maltophilia*, was more resistant to enzymatic cleavage and remained stable when exposed to murine bronchoalveolar lavage fluid (BALF) [162]. Importantly, the antibacterial activity of the peptide was conserved when the compound was tested against CF isolates of *P. aeruginosa* in 25% CF sputum in the presence of sodium chloride and DNase I. Under these conditions, D-BMAP18 prevented the deposition of new biofilm and eradicated preformed biofilms of *P. aeruginosa*. The in vitro anti-bacterial capacity of the compound was also coupled with anti-inflammatory properties [165]. However, a certain level of cytotoxicity was detected against human pulmonary epithelial cells and toxicity was observed in mice [162].

The synthesis of an aliphatic analog of D-BMAP18 (D-BMAP18_FL) did not reduce the cytotoxicity of the compound, but it showed increased anti-inflammatory properties and antibacterial and antibiofilm activities, similar to the aromatic form of the molecule [166].

A promising strategy to reduce toxicity, increase stability and maintain the in vivo efficacy of AMPs was proposed by Forde and colleagues [167]. The authors designed pro-peptides with an anionic *N*-terminal pro-moiety, able to mask the net positive charge of the molecule and containing a substrate for the enzyme neutrophil elastase (NE), released by neutrophils at the site of infection. The negatively charged pro-region sequesters the active peptide and masks its net positive charge—thus limiting its potential toxicity to endobronchial cells—and makes the activity of the resulting peptide dependent on the proteolytic activity of the host NE at the site of infection. 

When this approach was employed to design derivatives of different natural host defense peptides in their D-forms,—D-Bac8c2 from Bactenecin2,5 Leu [168], D-HB43 from crustacean polyphemusin I [169], D-P188 Leu from cecropin A–magainin 2 hybrid [170] and WMR, derived from innate immunity peptides found in the hagfish *Myxine glutinosa* [171],—the obtained pro-peptides could be activated by CF BALF in the presence of sodium chloride. The pro-AMP modification reduced host toxicity, increased specificity and, depending on the considered peptide, restored the antibacterial activity against *P. aeruginosa* in NE-rich CF human BALF under high salt conditions [167,172].

When the same strategy was applied to synthesize an inactive D-BMAP18 proform (Pro-D-BMAP18), the resulting compound could be efficiently converted to active D-BMAP18 by the NE and proteolytic enzymes present in CF sputum. The activated peptide retained the antimicrobial capacity of the parental form against *P. aeruginosa* and showed higher biocompatibility towards human cells [173]. 

Another approach to improve the stability of AMPs to proteolytic degradation is to synthetically produce them in the form of dendrimers, defined as hyperbranched polymeric molecules [174,175]. SET-M33, a non-natural cationic antimicrobial peptide built in branched form, displayed high resistance to degradation in biological fluids and antibacterial and anti-biofilm efficacy against multidrug- and extensively drug-resistant Gram-negative isolates, including *P. aeruginosa* in vitro and in vivo [176]. In preclinical infection models, the peptide exhibited reduced toxicity and anti-inflammatory activity. SET-M33 nanoparticles designed for pulmonary administration retained in vitro antibacterial activity against *P. aeruginosa* and displayed long pulmonary residence and efficacy in a *P. aeruginosa* lung infection mouse model [177]. The synthesis of a two-branched dimeric form of the peptide (SET-M33DIM) allowed for the reducing of cytotoxicity and inflammation but resulted in a molecule with decreased antibacterial activity [178]. 

G3KL and TNS18, two of the most potent peptide dendrimers reported to date, efficiently killed Gram-negative bacteria, including *P. aeruginosa* [179,180].

### 5.3. Antimicrobial Peptides in Clinical Trials for Cystic Fibrosis Infections

Naturally inspired or de novo engineering of AMPs allowed the design and synthesis of molecules that were able to reach clinical trials. WLBU2, a de novo engineered cationic amphipathic 24-residue peptide with membranolytic activity, is composed of Arg, Val, and Trp [181]. Thanks to its broad-spectrum antibacterial activity against ESKAPE (*Enterococcus faecium*, *Staphylococcus aureus*, *Klebsiella pneumoniae*, *Acinetobacter baumannii*, *Pseudomonas aeruginosa* and *Enterobacter* species) pathogens, this molecule is now in phase 1 clinical trials for the treatment of periprosthetic joint infections. Importantly, WLBU2 prevents *P. aeruginosa* biofilm formation on polarized human bronchial epithelial cells. Moreover, it is effective in reducing bacterial burden and *P. aeruginosa*-associated inflammation when intratracheally administered in a murine lung infection model [182]. As reported above, enantiomerization constitutes a useful approach for the optimization of antimicrobial peptides to improve their efficacy, stability or safety. The D-enantiomerization of WLBU2 led to higher activity against bacterial biofilms, reduced host toxicity and enhanced stability when delivered into the airway [183].

Inhalation therapy allows for achieving high drug concentrations in the lungs, thus maximizing efficacy while limiting systemic adverse effects. However, the limited number of available inhaled antibiotics (tobramycin, colistin and aztreonam) hinders the efficacy of long-term treatments. Murepavadin, also known as POL7080, is a *P. aeruginosa*-specific, peptidomimetic antibiotic which is currently moving to phase 2 clinical trials as an inhalation therapy for CF and non-CF bronchiectasis lung infection. It belongs to the novel class of outer membrane protein targeting antibiotics: by binding to LPS transport protein D (LptD), it inhibits LPS transport, leading to the alteration of the outer membrane and, ultimately, cell death. This mechanism of action allows potent activity against XDR (extensively drug-resistant) and CF isolates of *P. aeruginosa*, even under conditions that resemble the CF lung environment, as in the presence of pulmonary surfactant and in artificial sputum [184,185,186]. 

## 6. Nanoparticles

Nanoparticles (NPs), ranging from 1 to 100 nm, have various compositions, structures, and sizes conferring specific chemical and biological properties, allowing several applications in medical and environmental fields. Based on their nature, NPs have been divided into two different groups, i.e., organic, or carbon-based, and inorganic particles. Figure 3 reports a graphical representation of the described NPs, grouped according to their composition. The following paragraphs will discuss the studied NP formulations in correlation with the different Gram-negative CF pathogens. 

### 6.1. Organic Nanoparticles

Organic NPs come in diversified shapes and structures and are mainly formed by an external capsule composed of polymers or lipids; specifically, encapsulation can decrease off-target toxicity and control drug release in the tissue [187]. Moreover, an organic NP can be broken down naturally, easing its excretion from the body. 

#### 6.1.1. Smart Nano-Systems

Smart nano-systems, as described by Chen et al. [188], can react to specific internal or external stimuli or be target-specific, through the presence of small molecules or peptides. Wang et al. [189] produced a nano-system to carry specific reactive oxygen species (ROS)-responsive elements to treat *P. aeruginosa* in a mouse infection model. 4-(hydroxymethyl) phenylboronic acid pinacol ester-modified α-cyclodextrin (Oxi-αCD) was used to encapsulate the antibiotic moxifloxacin (MXF), originating a novel ROS-sensitive MXF/Oxi-αCD NP; an additional modified 1,2-distearoyl-sn-glycero-3-phosphoethanolamine-*N*-amino(polyethylene glycol) (DSPE–PEG)–folic acid polymeric coating was added to enable the active targeting of infected lung macrophages, in order to eliminate intracellular bacteria. In addition to displaying lower MIC values against different clinical *P. aeruginosa* strains compared to MXF alone, folic-acid modified MXF/Oxi-αCD NPs successfully decreased the bacterial content inside activated macrophages, while increasing mice survival rate up to 40% after six days of infection; in parallel, no significant cytotoxic effect was found on RAW264.7 and A549 cells [189]. 

Similarly, Pinto et al. [190] developed a NAC encapsulated lipid nanoparticle functionalized with D-phenylalanine, D-proline, and D-tyrosine; NAC exerted both mucolytic and antibiofilm activity, while D-amino acids can target and disassemble newly and already existing biofilms of *P. aeruginosa*. No cytotoxicity or hemolysis was highlighted after incubation with 2 mg/mL of loaded NPs, compared to the unloaded ones; nevertheless, the addition of NAC to the functionalized NPs only affected the bacterial viability if combined with different concentrations of MXF.

#### 6.1.2. Lipid-Based Nanoparticles

Lipid-based systems, such as liposomes and solid lipid nanoparticles (SLNs), are mainly composed of phospholipids and cholesterol, and differ in the organization of their lipid layer [187]. Liposomes present a lipid bilayer surrounding an aqueous pocket, which can contain hydrophobic or hydrophilic molecules or proteins, and usually produce few adverse effects. Depending on the external charge, liposome-encapsulated drugs would easily interact with bacteria, despite their low outer membrane permeability, as in the case of Bcc and *P. aeruginosa*. 

As a proof of concept, Messiaen et al. [191] studied the bactericidal activity of tobramycin-loaded negatively charged liposomes against different Bcc biofilms. Here, an anionic liposomal formulation with tobramycin showed a bactericidal effect only against *B. cepacia* LMG 1222, compared to free tobramycin being effective against all the tested strains, while demonstrating liposome enrichment close to biofilm clusters. Such a difference may be attributable to the electrostatic repulsive forces limiting the fusion of negatively charged liposomes with the LPS-rich outer membrane; neutral and positively charged coating did not produce any improvement. 

Lipid coating is functional also for non-antibiotic entities, such as prophage lysins [192]; from a set of 19 proteins identified from 38 prophages within different *P. aeruginosa* genomes, lysins Pa7 and Pa119 were found to be biologically active against *P. aeruginosa* cells once encapsulated in dipalmitoylphosphatidylcholine:dioleoylphosphatidylethanolamine:cholesteryl hemisuccinate (DPPC:DOPE:CHEMS) liposomes. Free lysins showed a bactericidal effect at 25 µg/mL, while their encapsulated counterparts were already active at 6.25 µg/mL; interestingly, the time needed for the treatment to be effective seemed to be dependent on lysin release and accumulation at the peptidoglycan site. 

In contrast, SLNs have a single phospholipid outer layer encapsulating both aqueous and non-aqueous content, making them easier to manufacture at scale and more stable from a kinetic point of view [193]. Sodium colistimethate-loaded lipid nanoparticles, i.e., colistin-SLNs and -nanostructured lipid carriers (NLCs) developed by Pastor et al. [194], allowed high drug entrapment and a sustained drug release profile, making them a valuable alternative to treat *P. aeruginosa* infections associated with CF; while their rigidity enabled an efficient nebulization for lung localization, no variation in the MICs was observed compared to the free drug. When colistin-SLNs’ IC_50_ was assayed on H441 and A549 human lung cells, an approximately 160- and 28-fold decrease in the cytotoxicity was highlighted compared to the uncoated colistin, with the less toxic formulation being the colistin-NLCs one. Moreover, colistin-NLCs’ suitable release profile was confirmed in vivo using a mouse model, as nebulization enabled a homogeneous delivery throughout the lungs and 48 h of persistence after administration. 

The promising properties of NLCs were studied by Vairo et al. [195] too, by encapsulating sodium colistimethate (SCM) in positively and negatively charged carriers in the presence of trehalose or dextran as a cryoprotectant. Positive charge polarization was achieved with a further chitosan coating. Release profiles were independent of the charge or cryoprotectant; compared to free colistin, positively charged SCM-NLCs exerted a relevant decrease in MIC, MBIC and MBEC when tested against different strains of *P. aeruginosa*.

Both the previous studies are somewhat concordant on the efficacy of nanostructured lipid carrier formulations, as they are characterized by an imperfect crystal structure, which allows a higher drug loading and a finer and more controllable drug release.

#### 6.1.3. Polymeric Nanoparticles

Polymeric nanoparticles (PNPs) are formed by small homogeneous molecules, which can be classified as natural or synthetic [188]; along with their small size and high drug encapsulation, they can easily diffuse through capillaries, making them the most studied platform for pulmonary delivery. The addition of a polymer, as highlighted earlier, is essential to modify nanoparticles’ surface net charge, facilitating drug diffusion.

Conversely, NPs containing bioactive natural polymers were studied by Montoya-Hinojosa et al. [196] and Patel et al. [197], by producing two different formulations containing chitosan. The first group encapsulated curcumin–chitosan–sodium tripolyphosphate (Cur–Chi–TPP) inside iron oxide magnetic nanoparticles (MNPs) and tested its efficacy in combination with trimethoprim–sulfamethoxazole (TMP–SXT) on different CF bacteria, i.e., Bcc, *A. xylosoxidans* and *S. maltophilia*, susceptible or resistant to TMP–SXT. Even if the presence of Cur–Chi–TPP MNPs had a lower antimicrobial activity on planktonic cells compared to CHI–TPP, the MBIC values were significantly lower for all the strains tested; low biofilm eradication activity was also observed for the three strains.

Patel et al. [197] functionalized chitosan nanoparticles of ciprofloxacin (CIP–CH NPs) by immobilizing alginate lyase (AgLase) to treat mucoid *P. aeruginosa* infection in CF. Again, MIC values of AgLase–CIP–CH NPs were unchanged, compared to the CIPR treatment; on the other hand, AgLase–CIP–CH NPs’ MBEC value at 24 h was four-fold lower, as a higher biofilm penetration could be attributed to the formulation. In addition, repeated doses of the treatment reduced the microbial load completely and disassembled the biofilm extracellular matrix; both biomass and thickness were significantly reduced. Nanoformulation did not display cytotoxicity against lung epithelial cells in dimethylthiazol (MTT) assays. 

In contrast to natural polymers, the synthetic ones come in different fashions and forms. Poly-lactic-co-glycolic acid (PLGA) is surely one of the most common FDA-approved co-polymers [188]. PLGA has been successfully employed to encapsulate antibiotics and antimicrobial peptides, with the best results obtained for the latter ones, as reported by Cresti et al. [198]. Compared to SET-M33 alone, the PLGA–PEG complexed SET-M33 peptide showed an increased diffusion rate through artificial mucus and bacterial alginate (Section 5.2). The PLGA-encapsulated peptide was active on both planktonic and sessile *P. aeruginosa* cells, while being non-toxic neither on 16HBE14o- and CFBE41o- cells, nor on mice. 

Eventually, diverse delivery systems can be combined as in the case of tobramycin liposomes embedded in different synthetic co-polymeric matrices (TOB MO-LCNPs) [199]. Such NPs allowed for the rapid release of tobramycin and maintained their structure after nebulization. A 100,000-fold decrease, up to a near-total eradication, in the load of *P. aeruginosa* PAO1 was highlighted when biofilms were exposed to increasing concentrations of formulated tobramycin compared to the free drug, whereas an increased penetration in the lower matrix layers was observed too. No signs of toxicity were evident in CFBE41o- cells. 

Another co-polymeric matrix, DSPG–PEG–OMe, was used as a stabilizer in an ivacaftor–colistin nanosuspension [200], thus combining an antibiotic with a CFTR potentiator. Thanks to its quinolone ring, ivacaftor itself can inhibit *P. aeruginosa* growth [201]. The compound showed minimal toxicity, despite the presence of colistin, and exerted potent bactericidal activity against *P. aeruginosa* compared to colistin alone. 

Another example of PNP application is provided by the work by Costabile et al. [202] which managed to increase the solubility of the FtsZ-inhibitor C109 in a D-α-tocopheryl polyethylene glycol 1000 succinate inhalable formulation, embedded in hydroxypropyl-β-cyclodextrin. While no decrease in the MIC values was observed against a panel of 10 *B. cenocepacia* strains, increased inhibitory activity against *B. cenocepacia* biofilm was evident, as well as an increase in larvae survival upon infection with *B. cenocepacia*, when the formulation was combined with piperacillin.

Another group of polymer-based NPs is represented by nanogels—three-dimensional hydrogel materials formed by cross-linked polymer networks and with a high capacity to hold water [203]. A formulation of this type was developed by Li et al. [204], by cross-linking chitosan–glutaraldehyde monomers to treat Bcc infections; the compound possessed increased antibacterial activity compared to chitosan and glutaraldehyde alone, with variable inhibition diameters depending on the strain. 

### 6.2. Inorganic Nanoparticles

In contrast to organic ones, inorganic NPs are mainly composed of metal or anionic ions, and thanks to their well-known antimicrobial properties they can be efficiently coupled with antibiotics [205]; unfortunately, an important drawback that has to be considered is their variable cytotoxicity, which can be reduced by coupling particles to liposomes or polymers.

Ionic NPs’ activity relies on their tendency to produce highly ROS and to interact with negatively charged LPS, allowing NPs entry with the consequent inhibition of protein synthesis and DNA damage. Ionic nanoparticle synthesis requires a highly reducing environment, either synthetically or naturally reproduced. Among the others, silver nanoparticles (Ag NPs) have been largely studied for their wide antimicrobial activity, as they can cluster along the cell wall and precisely at the poles, leading to membrane blebbing [206]. Compared to the other ionic NPs, they present low cytotoxicity. 

Pompilio et al. [207] developed an electrochemically-produced silver-based formulation to respond to the increasing MDR bacteria associated with CF; these Ag NPs showed lower or still comparable MIC and MBC values against different strains of *P. aeruginosa*, compared to tobramycin, with the more significant results against strains of *B. cepacia* and *S. maltophilia*; no toxicity was reported in *G. mellonella*. 

Green silver NPs were produced by Al-Momani et al. [208], using leaf and fungi extracts. When tested against a panel of six *P. aeruginosa* strains isolated from pwCF, sub-inhibitory concentrations significantly affected growth and biofilm formation, as well as the expression of QS genes [208].

Ag NPs have proved to be effective, also, in treating *H. influenzae* and carbapenem-resistant *P. aeruginosa* isolates [209].

The effect of phytofabricated Ag NPs prepared using *Cuphea carthagenensis* was evaluated on biofilm, QS and QS-dependent virulence factors of *P. aeruginosa*, leading to a significant attenuation of each trait at sub-MIC concentrations [210]. Moreover, they did not impact planktonic bacteria growth and were not toxic in human cell lines [210]. 

Other metallic NPs include selenium ones, effective in inhibiting *H. influenzae*, *B. cenocepacia*, *P. aeruginosa* and *S. maltophilia* [209,211]. Antibiofilm activity was confirmed for the last two bacteria [211]. 

Gold NPs functionalized with NAC or chitosan oligosaccharide interfered with newly and already formed *P. aeruginosa* biofilms [212,213], by either reducing viscosity or hampering QS-dependent virulence factors, even at very low concentrations [213]. 

A similar effect was obtained with samarium-oxide NPs developed by Zahmatkesh et al. [214], against MDR *P. aeruginosa*. While growth was inhibited at 50 µg/mL, concentrations below the MIC disrupted pyocyanin production and motility and reduced proteolytic and hemolytic activities, in addition to biofilm formation. 

Finally, an example of an anionic biocompatible antimicrobial formulation consists of colistin-loaded calcium phosphate NPs. Developed by Iafisco et al. [215], it provided a safe and efficient delivery system against *P. aeruginosa* RP73, suitable for pulmonary administration and characterized by a high colistin payload, without impairing activity, diffusion through mucin, and antibiofilm activity. 

## 7. Discussion and Conclusions

Antibiotic therapy remains one of the pillars of cystic fibrosis (CF) lung disease management. The recent introduction of CFTR modulator therapies resulted in improved innate defense mechanisms in most patients’ airways, potentially leading to the clearance of the pre-existing lung infections. However, many reports are now underlying the persistence of the chronic bacterial infections in modulator treated people, proving that they still represent an unsolved serious problem [216]. The continuous administration of inhaled colistin, tobramycin, levofloxacin and aztreonam is widely used as a maintenance treatment against opportunistic Gram-negative lung infections, in particular after the establishment of *P. aeruginosa* chronic infections [216]. However, the increasing isolation of MDR strains, beside the intrinsic antibiotic resistance of CF pathogens such as *P. aeruginosa*, *B. cenocepacia*, *S. maltophilia*, *A. xylosoxidans* and *H. influenzae*, make these treatments often ineffective, even when used in combination [217]. The critical shortage of new antibiotics requires the development of novel strategies to give new life to outdated or Gram-negative spectrum antimicrobials, in order to broaden the therapeutic options available in clinics.

In this review, we described different alternative approaches to the use of antibiotics which have been investigated in the last five years, especially for the treatment of the above-mentioned Gram-negatives.

First of all, we described anti-virulence compounds: in this case, the goal is to block bacterial traits that allow the establishment of a successful infection, including the QS pathway (which regulates the expression of many virulence factors itself), the ability to form biofilms, the activity of metalloproteases, siderophores or secretion systems. This strategy renders the insurgence of drug resistance less probable, since it does not affect bacterial viability. Regarding *P. aeruginosa*, there are a lot of results about molecules being able to interfere with virulence factor expression and biofilm formation, while only a few anti-virulence compounds that hit the other CF Gram-negative pathogens have been described. Only a small number of compounds are fully characterized; hence, it is necessary to proceed in this investigation field to obtain compounds which can enter clinical trials. However, until now, these molecules can be used only in combination with antibiotics, since their efficacy is not fully characterized.

A second strategy is the use of phage therapy: here, the ability of bacteriophages to selectively kill specific bacteria is exploited, to bypass the issue of the accumulation of mutations in antibiotic encoding targets, as well as the activation of efflux pumps, modifying enzymes, issues about permeability, and so on. On the other hand, the range of activity of phages can be broadened using cocktails. Among the main concerns about the use of phages, there are the shortage of controls, criticisms about the variability of achieved results, their clinical safety and dose optimization [218]. Moreover, undesired impacts on the human microbiome could occur because of the wide distribution of phages in the environment, while lytic phages (the preferred ones for phage therapy) may be able to invade eukaryotic cells and induce innate immune responses [219]. Another concern, which is particularly important in the case of Gram-negative bacteria, is the possible release of endotoxins during therapy [220]. All these challenges need to be addressed before phage therapy can be successfully implemented. 

Another valuable alternative approach to the use of the antibiotics alone is the administration of adjuvants. These molecules have the ability to impair bacterial resistance mechanisms, thus exerting a synergistic effect with the antibiotics, potentially shortening the duration of the treatment, reducing toxicity and delaying the selection of resistant mutants [221]. However, in this case, further studies about toxicological safety in vitro and in vivo are lacking and should be implemented before their effective introduction in clinical practice. 

Regarding antimicrobial peptides, the ones with membranolytic activity have attracted attention for their clear advantages, compared to traditional antibiotics. Their good antimicrobial activity against a broad range of drug-resistant bacteria is exerted through a rapid mechanism of action that confers a limited propensity to generate resistance [222]. Moreover, synergistic effects, upon co-administration with conventional antibiotics, have been described [223]. Importantly, not only they can inhibit bacterial planktonic growth, but AMPs with broad antibiofilm activity have been reported [224]. These properties, together with their anti-inflammatory and immunomodulatory effects, make AMPs an attractive tool to fight Gram-negative bacteria, especially those responsible for chronic infections in pwCF [225]. However, although AMPs represent an appealing alternative to the use of antibiotics in the fight against bacterial infections in pwCF, it is important to consider that the sensitivity to the antibacterial action of these peptides depends on the considered bacterial species. Notably, *Burkholderia* sp. display an intrinsic resistance to cationic AMPs. This is mainly due to the unique properties of their outer membrane lipopolysaccharide, including the constitutive, essential modification of outer membrane Lipid A phosphate groups with cationic 4-amino-4-deoxy-arabinose. By decreasing the overall negative charge on the outer membrane LPS, this modification leads to a reduction in the binding, accumulation and permeation of cationic AMPs [226]. Finally, in vivo experiments are generally performed using murine models of *P. aeruginosa* infection. Consequently, the in vivo activity of AMPs against other CF pathogens is poorly understood. 

The last strategy we described is the use of nanoparticle-based formulations to overcome Gram-negative MDR infections associated with CF. Their increased bioavailability, stability and ability to diffuse through biofilms make them a suitable therapeutic option [227]. Once a particular antimicrobial is loaded into the nanoparticle, its size, surface area and highly reactive nature allow for a clear improvement in safety, accumulation at the infection site and toxicity against the pathogen, compared to the unloaded compound. Some aspects to consider before their application include: the evaluation of their compatibility with blood [228], eventual toxicity due to their accumulation in the spleen and liver [229], the final application, the size, biocompatibility, biodegradability and the capability of encapsulation of the antibiotic in order to select the most appropriated polymer to use [230]. Also, the surface of the nanoparticle should allow the bypassing of the human immune system [231]. Another issue is the insurgence of inflammation associated with ROS generation, which depends on the surface coating, dissolution and other physical characteristics of the nanoparticle [232].

Overall, the costs associated with these new therapies are quite high and, as reported for all the different technologies, further studies are necessary to fully establish the possibility of introducing them in clinical practice.

## Figures and Tables

**Figure 1 antibiotics-13-00071-f001:**
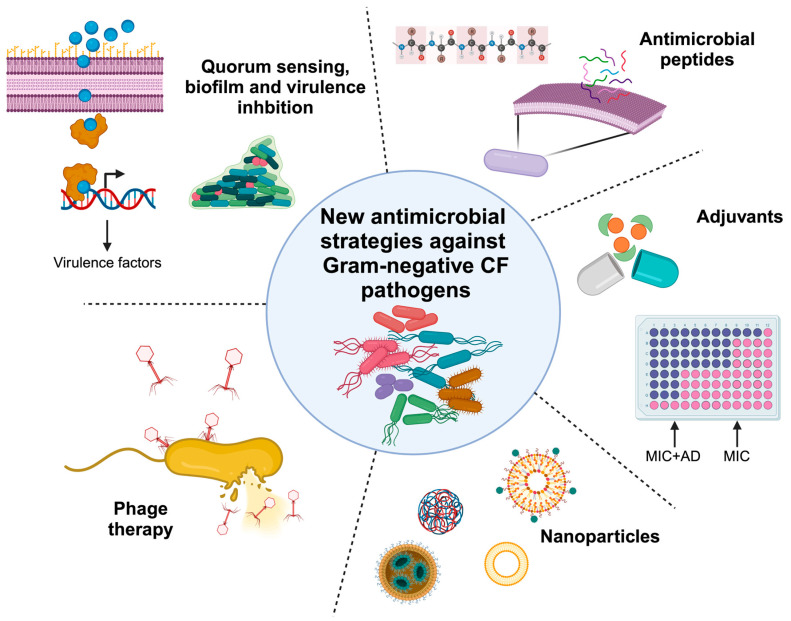
Alternative strategies to cope with multi-drug resistant CF pathogens described in this review. MIC: minimum inhibitory concentration; AD: adjuvant. Created with Biorender.

**Figure 2 antibiotics-13-00071-f002:**
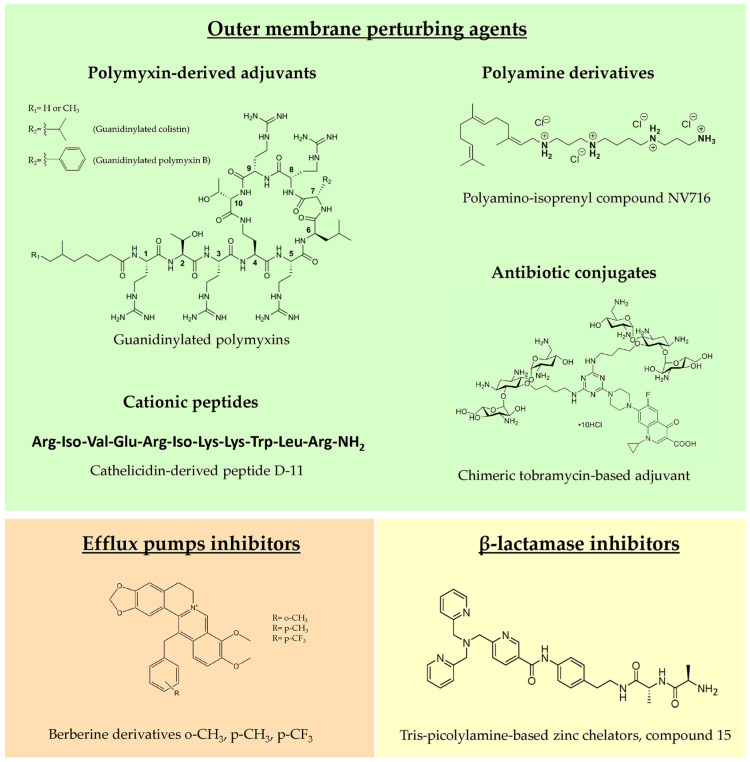
Representative examples of novel molecules belonging to the three main classes of antibiotic adjuvants.

**Figure 3 antibiotics-13-00071-f003:**
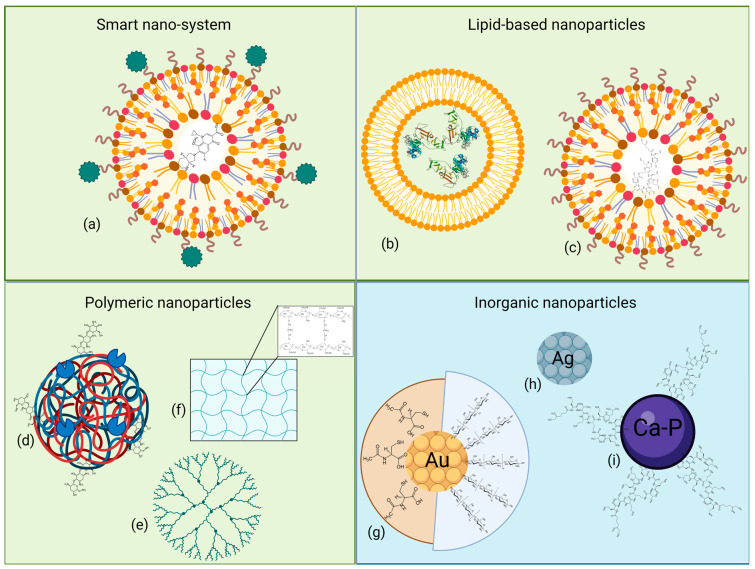
Graphical representation of some of the described NPs, grouped according to their chemical nature and structure. (**a**) Smart nanocarrier system coated with responsive elements; (**b**) liposome; (**c**) solid lipid NP; (**d**) heteropolymeric NP complexed with enzymes and antibiotics; (**e**) dendrimer; (**f**) hydrogel; (**g**) gold, (**h**) silver and (**i**) calcium–phosphate NPs, with or without linked bioactive compounds. Light green background indicates organic origin, while light blue background indicates inorganic one. Created with Biorender.

**Table 1 antibiotics-13-00071-t001:** Synthetic molecules inhibiting quorum sensing.

Synthetic Molecules			
Quorum Sensing Inhibitors	Bacteria	Mechanism	Reference
AHL analogs	*P. aeruginosa*	3O-C12-HSL competition	[7]
Halogenated furanone derivatives	*P. aeruginosa*	LasR and RhlR inhibition	[8]
Halogenated furanone derivatives with palladium-catalyzed coupling reactions	*P. aeruginosa*	LasR and RhlR inhibition	[9]
PqsR modulators	*P. aeruginosa*	PqsR inhibition	[10]
3-hydroxypyridin-4[1H]-one derivatives	*P. aeruginosa*	*pqs* inhibition	[11]
Small QS inhibitor, squalene derived nanoparticles formulation	*P. aeruginosa*	unknown	[12]
Nitrofurazone and Erythromycin estolate	*P. aeruginosa*	unknown	[13]
Niclosamide	*P. aeruginosa*	3OC12-HSL signaling process	[15]
Clofoctol	*P. aeruginosa*	PqsR competitive inhibitor	[15]
Tyramine	*B. cenocepacia*	CepI/R and CciI/R inhibition	[18]
Diketopiperazines	*B. cenocepacia*	CepI inhibition	[19,20]
Sulfonamide-based DSF bioisosteres	*S. maltophilia* and Bcc	DSF inhibition	[21]

**Table 3 antibiotics-13-00071-t003:** Synthetic anti-virulence molecules.

Synthetic Molecules			
Anti-Virulence Molecules	Bacteria	Target	Reference
Psammaplin A and bisaprasin	*P. aeruginosa*	LasB	[34]
Benzoxazolone derivatives	*P. aeruginosa*	Pyocyanin	[35]
Gallium nitrate	*P. aeruginosa*	Siderophores	[36]
MEDI3902	*P. aeruginosa*	PcrV and Psl	[37]
Fluorothiazinon	*P. aeruginosa*	T3SS	[38]
Nonpeptidic inhibitors	*H. influenzae*	IgA1 proteases	[39]
FR90098	*B. cenocepacia*	non-mevalonate pathway	[40]

**Table 4 antibiotics-13-00071-t004:** Natural compounds with anti-quorum sensing or antibiofilm activity.

Natural Compounds		
Quorum Sensing Inhibitors	Bacteria	Reference
Coumarin and coumarin derivatives	*P. aeruginosa*	[41,42]
Baicalin	*P. aeruginosa*, *Bcc*	[43,44]
Oridonin	*B. cenocepacia*, *Burkholderia species*	[46]
*Chromohalobacter* sp. D23	*B. cepacia*	[47]
Celastrol	*S. maltophilia*	[48]
**Biofilm inhibitors**	**Bacteria**	**Reference**
*Dioon spinulosum* extract	*P. aeruginosa*	[49]
Plant-derived triterpenes (analogs of oleanolic acid)	*P. aeruginosa*, *B. cenocepacia*	[50]
Plant essential oils	*P. aeruginosa*, *H. influenzae*	[51,52,53,54]
Ceragenins (CSAs)	*A. xylosoxidans*	[55]
Metalloproteases from stony coral	*S. maltophilia*	[56]
Vitamin C	*S. maltophilia*	[57]
*Allium stipitatum* extract	*S. maltophilia*	[58]
Glycosyl hydrolases PslG from *P. fluorescens*	*Pseudomonas strains*	[59]

## Data Availability

No new data were created or analyzed in this study. Data sharing is not applicable to this article.

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
