# Peer review of "New Antimicrobial Strategies to Treat Multi-Drug Resistant Infections Caused by Gram-Negatives in Cystic Fibrosis"

_antibiotics, 2024, doi:10.3390/antibiotics13010071_

Round 1

Reviewer 1 Report

Comments and Suggestions for Authors

The manuscript from Scoffone et al. reviews the novel antimicrobial strategies to treat MDR infections caused by Gram-negatives in CF, focusing in anti-virulence and anti-biofilm compounds, phage therapies, antibiotic adjuvants, antimicrobial peptides and nanoparticle formulations. I only have minor corrections to suggest.

Minor revisions:

1)      Introduction, 1st paragraph: It is missing references.

2)      Figure 1: Please correct “Gram-negatives” to “Gram-negative”.

3)      Figure 1: The abbreviations of the figure “MIC” and “AD” should be described in the legend.

4)      Table 1: Please confirm if the last row the bacteria is “B. cepacia” or “Bcc”, because in the manuscript reference 18, they refer to Bcc and not only B. cepacia.

5)      Table 3: The IGA1 protease inhibitors don’t have a specific name?

6)      Table 4, row 10: Please correct “cenocapcia” to “cenocepacia”.

7)      Discussion, line 992: Please confirm if the phrase is correct “…Gram-positive spectrum antimicrobials…”

8)      Abbreviations missing full name description: FDA, PQS, pwCF.

9)      Reference 8 incomplete.

10)   Reference 46: Title incomplete.

11)   Reference 118 and 119: different format from the other references.

12)   Reference 146: is out of order in reference list.

Author Response

The manuscript from Scoffone et al. reviews the novel antimicrobial strategies to treat MDR infections caused by Gram-negatives in CF, focusing in anti-virulence and anti-biofilm compounds, phage therapies, antibiotic adjuvants, antimicrobial peptides and nanoparticle formulations. I only have minor corrections to suggest.

We thank the reviewers for his/her comments and corrections.

Minor revisions:

  • Introduction, 1st paragraph: It is missing references.

Two additional references were included:

  • Françoise, A., Héry-Arnaud, G. (2020). The Microbiome in Cystic Fibrosis Pulmonary Disease. Genes11(5), 536.
  • Perikleous, E. P., Gkentzi, D., Bertzouanis, A., Paraskakis, E., Sovtic, A., & Fouzas, S. (2023). Antibiotic Resistance in Patients with Cystic Fibrosis: Past, Present, and Future. Antibiotics (Basel, Switzerland)12(2), 217.

Please note that, due to the introduction of these references, all the reference numbers have been changed. However, the changes are tracked only in the text because in the reference paragraph the overwriting was too confusing.

2)      Figure 1: Please correct “Gram-negatives” to “Gram-negative”.

            Done as requested.

3)      Figure 1: The abbreviations of the figure “MIC” and “AD” should be described in the legend.

Done as requested.

4)      Table 1: Please confirm if the last row the bacteria is “B. cepacia” or “Bcc”, because in the manuscript reference 18, they refer to Bcc and not only B. cepacia.

            The table was modified as suggested.

5)      Table 3: The IGA1 protease inhibitors don’t have a specific name?

            We thank the referee for the comment. These inhibitors were identified using a high-throughput screening and they are characterized by different chemical groups: an ester with limited aqueous half-life (compound 1), and two carbamates that are stable in aqueous solution (compounds 2 and 3) (Shehaj et al., 2019). We modified the name in the Table 3, indicating these compounds as “nonpeptidic-inhibitors” since these molecules are the first IgA1 protease inhibitors of this type.

6)      Table 4, row 10: Please correct “cenocapcia” to “cenocepacia”.

The typo was corrected.

7)      Discussion, line 992: Please confirm if the phrase is correct “…Gram-positive spectrum antimicrobials…”

We are sorry for the mistake; the sentence has been changed to “… Gram-negative spectrum antimicrobials…”.

8)      Abbreviations missing full name description: FDA, PQS, pwCF.

The full name description of each abbreviation was reporter upon first use.

9)     Reference 8 incomplete.

10)   Reference 46: Title incomplete.

11)   Reference 118 and 119: different format from the other references.

12)   Reference 146: is out of order in reference list.

9-12) We apologize for the oversight. All references have been corrected and double-checked.

Reviewer 2 Report

Comments and Suggestions for Authors

This review discusses all the alternative approaches to fight MDR Gram-negative CF pathogens. Overall, it's a very informative and comprehensive review that would benefit this field. The manuscript is well organized and easy to understand. I recommend to accept this manuscript for publication.

Minor:  please add more introduction information regarding the causes, mechanisms and consequences of infection in cystic fibrosis.

Author Response

This review discusses all the alternative approaches to fight MDR Gram-negative CF pathogens. Overall, it's a very informative and comprehensive review that would benefit this field. The manuscript is well organized and easy to understand. I recommend to accept this manuscript for publication.

We thank the reviewer for his/her kind appreciation of our work.

Minor:  please add more introduction information regarding the causes, mechanisms and consequences of infection in cystic fibrosis.

We thank the reviewer for the suggestion. We have included the requested information in an additional paragraph in the introduction section (lines 23- 34).